# Experimental Investigation on Friction and Wear Behavior of the Vertical Spindle and V-belt of a Cotton Picker

**DOI:** 10.3390/ma12050773

**Published:** 2019-03-06

**Authors:** Auezhan Amanov, Jan Putra Bahtra Agung Sembiring, Tileubay Amanov

**Affiliations:** 1Department of Mechanical Engineering, Sun Moon University, Asan 31460, Korea; janpoetra22@gmail.com; 2Institute of Mechanics and Seismic Stability of Structures, Academy of Sciences, Tashkent 100125, Uzbekistan; amanov43@mail.ru

**Keywords:** ST3, roughness, hardness, friction, slip, wear resistance, cotton picker

## Abstract

This study deals with the friction and wear behavior of the vertical spindle and V-belt to improve the reliability, operation and to extend the service life of a cotton picker. The vertical spindle made of low-carbon steel (ST3) was treated by the ultrasonic nanocrystal surface modification (UNSM) technique to control the friction and wear behavior. It was found that the UNSM technique reduced surface roughness and increased surface hardness of the vertical spindle. The friction and wear behavior of the vertical spindle and V-belt was assessed by carrying out tribological tests and the results showed that the UNSM-treated vertical spindle generated a higher friction coefficient compared to the untreated one due to having less slip. In case of wear resistance, unmeasurable wear occurred on the surface of the vertical spindle due to its significant high hardness compared to the hardness of the V-belt that came into contact with the vertical spindle in relative motion. Hence, the wear behavior and mechanisms of the V-belts were systematically investigated and also discussed based on the wear track profiles and micrographs. It can be concluded that the application of the UNSM technique to the vertical spindle may contribute to improve the performance of cotton pickers by reducing the slip and prolonging the service life.

## 1. Introduction

It is well known that vertical spindle drums of cotton pickers consist of a gland cylinder with protrusions on the outer surface and spindles. In the working area during the rotation of the drum, cotton is captured by spindles and pulled out of the boxes. The pressing cylinder serves to fix the cotton on the spindles. However, the lower part of the spindle does not capture all the cotton from the bolls during the work, as it accounts for the first collection of a large mass of cotton and uncovered cotton tends to fall to the ground. The protrusions of the gland cylinder are made along a helix in order to increase the completeness of the cotton harvest in the drum [1,2].

The belt transmission driving mechanism is widely used to transmit power and motion in vertical spindle drums of cotton pickers. The reason for this is that this driving system has some benefits such as the possibility of decreasing and increasing the rotating speed of the vertical spindle with many possible speed ratios, high efficiency, cleanliness and low noise, and it also does not require lubrication [3]. One of the implementations of this mechanism can be found in vertical spindle drums of cotton pickers for transmitting motion from the actuator to the several vertical spindles around the main shaft of the actuator. The main shaft motion is transmitted to the vertical spindles through contact between the V-belt and grooves of the vertical spindle sheaves. The transmission occurs through friction between the V-belt and vertical spindle sheaves as a result of the relative slip motion [4,5]. The vertical spindle sheaves and V-belt contact will create a sliding friction between two surfaces, and this leads to the presence of abrasive wear. The friction and wear of the vertical spindle sheaves and belt will affect the performance of the cotton pickers by providing slips during transmission [6,7]. Slippage occurs in the belt transmission system when there is not enough belt tension and grip between the vertical spindle and V-belt [8]. Slip also will reduce the efficiency of transmission and results in simultaneously reducing the speed of vertical spindle drums of cotton pickers during the transmission and increasing the power loss. Besides slippage, the worn-out vertical spindle will also shorten the V-belt life and deteriorate the surface of V-belt, which leads to a noisy operation [9].

To improve the performance of cotton pickers, control of the friction coefficient and wear resistance of the vertical spindle is required. It can be achieved by modifying the surface integrity and the mechanical properties of vertical spindle drums of cotton pickers through an induced severe plastic deformation (SPD) method. One of the methods for inducing SPD is known as the ultrasonic nanocrystal surface modification (UNSM) technique, which utilizes ultrasonic energy to induce the SPD and to strengthen the materials by refining the coarse grains into nano-size grains with a certain depth from the top surface simultaneously. It has already been reported in the literature that the UNSM technique is capable of controlling the friction and wear behavior of various metallic materials by improving the surface integrity and increasing the mechanical properties [10,11,12,13,14,15]. In addition, it is well documented that the UNSM technique can enhance the wear resistance because of the increase in hardness due to the presence of a nanocrystal layer with nano-grains, while the control of friction behavior is mainly attributed to the change in surface integrity [16,17].

In this study, the vertical spindle of cotton picker made of low-carbon steel (ST3) was treated by the UNSM technique. The objective of this study is to investigate the effectiveness of the UNSM technique on the friction coefficient and wear resistance of the vertical spindle in comparison with the untreated one that comes into contact with the V-belt. The investigation concerns the surface integrity, mechanical properties and also the dimensions of wear tracks of the untreated and UNSM-treated vertical spindles of the cotton picker. It is expected that the control in friction and wear behavior between the V-belt and vertical spindle may result in better performance of the cotton picker by reducing the slip and prolonging the service life. A brief overview of cotton picker and UNSM technique is provided in the following sections.

## 2. Experimental Procedure

### 2.1. Specimen Preparation

In this study, a vertical spindle made of ST3 with a hardness of 23 HRC was cut into half sheave specimens by wire cutting. The cutting was performed to fit the sheave on the tribo tester during friction test. The dimensions of the vertical spindle are illustrated in Figure 1a. The dimensions of actual two different V-belts that come into contact with the vertical spindle of cotton picker are also illustrated in Figure 1b. The shore hardness, chemical composition and types of two different belts are listed in Table 1. Construction of both belt materials is completed using rubber and both are not reinforced. The mechanical properties and chemical composition of the vertical spindle are listed in Table 2 and Table 3, respectively. The distribution of chemical composition of ST3 is also shown in Figure 2, where detected oxide is a result of careless sample storage. Following the wire cutting, a series of vertical spindles were subjected to the UNSM technique, where the optimized treatment parameters are listed in the following sub-section. The UNSM treatment area was the inner part of a groove which is the contact point with the V-belt (see Figure 1a). It needs to be mentioned here that no polishing was done to the vertical spindles before and after UNSM treatment.

### 2.2. The UNSM Technique

The UNSM technique is one of the most simple surface modification technologies, which utilizes ultrasonic energy. During the treatment, both the static and dynamic loads are used to introduce an SPD by striking the surface with a certain amplitude up to 100 µm at a high-frequency of 20 kHz at room and high temperatures (RT and HT). Tungsten carbide (WC) ball is used to strike the surface of the vertical spindle. A series of vertical spindles were subjected to UNSM technique and the optimized treatment parameters used in this study are listed in Table 4. The optimization of UNSM technique parameters was made based on the reduction in surface roughness and increase surface hardness. It needs to be noted here that the V-belts were not subjected to the UNSM technique.

### 2.3. Working Principle of the Cotton Picker

The cotton picker is usually attached to the frame of the machine with the help of racks and axles. It consists of a permanent magnet of a spindle roller hardened by an element with a hysteresis loop. This loop is displaced by a nut fixed by means of a non-magnetic bracket on the surface of the spindle drum opposite the spindle roller. The non-magnetic bracket is under the test on the radius connecting the center of the drum with the spindle under test roller between the planes of the coil frame ends of the two half housing bodies. The apparatus is driven from the transfer reducer through the cardan shafts and the reducer is moved further through the gearing system. The rotation is transmitted to the spindle drums and pullers. The spindle drum is designed to capture raw cotton from the opened boxes in the working chamber and transport it to the pickup area. The drum consists of a shaft with a lower disk on which the fingers of spindles are fixed. On the shaft the upper disk is mounted in the holes, which are inserted into vertical spindles. The vertical spindles are designed to capture raw cotton from the boxes. The vertical spindle is a rod in the lower end of which the sleeve is pressed, and in the upper three-strand roller on which the ball-bearing of the closed type is mounted. On the rod wearing an exciting element in the form of a toothed tape. The drive of the vertical spindles is designed to rotate the spindles in the zone collecting the tip of the teeth of the tape forward, in the zone of removal in the opposite direction. The spindles are driven by three V-belts connected by clips with hooks, and on the other they are connected to the hook via springs, which create a constant clamping of the belts to the rollers as shown in Figure 3. The clutches are mounted so that the rollers run onto the belts from the side. The vertical spindle is rubbed against belt which is coupled with main shaft. The main shaft is connected to an actuator and spins at a speed of 110 rpm. This rotation will be transmitted to the spindle through contact between the V-belt and spindle, which makes the spindle also spin at a speed of 1200 rpm in result [18]. The contact between the vertical spindle and V-belt while rubbing against each other result in an unstable friction between the surface of spindle and V-belt, leading to a reduction in efficiency and an increase in power loss. Due to the unstable friction and impact, a wear will emerge on both surfaces that provide a malfunction of drums and shorten the service life. In this regard, a stable friction with stick-slip elimination is needed to control the friction and wear behavior.

### 2.4. Tribological Test

While using the actual vertical spindle as specimens, it is expected that the actual friction behavior can be simulated by tribological test. To study and evaluate the effects of the UNSM technique on friction and wear behavior, the tribological tests were performed. The selected tribological test conditions are listed in Table 5. The tribological tests were performed by rubbing two different V-belts against the untreated and UNSM-treated vertical spindle under dry conditions. Through this test, the friction behavior of the vertical spindle and V-belt was evaluated in terms of the friction coefficient with respect to sliding cycles and wear resistance. The friction coefficient was obtained as the ratio of tangential force (*F*) to the normal load (*N*), where both were measured using a sensor. In this tribological test, the actual vertical spindle and V-belt contact mechanism was simulated using a tribo tester (S&G, Incheon, Korea), where the actual V-belt was cut and glued to the pin as shown in Figure 4a. In this study, two V-belts made of different polymers were used as a counterface to compare the response of the V-belts while rubbing against the identical vertical spindle. The V-belts were cut to be fixed on the pin as shown in Figure 4b.

### 2.5. Surface Characterizations

All of the vertical spindles were washed before and after UNSM treatment using an ultrasonic bath (SD-80H, Mujigae, Seoul, Korea) in acetone (CH_3_)_2_CO for 10 min for deliverance of impurities. The surface roughness of the vertical spindle was measured using a two-dimensional surface profilometer (Mitutoyo: SJ-210, Kawasaki, Japan). The same profilometer was also used to obtain a surface profiling of the wear track formed on the surface of the V-belts. The surface hardness was measured using a portable hardness tester (GE Inspection Technologies: MIC 10, Lewistown, PA, USA). Both measurements were repeated at least 3 times in order to achieve a reliable result and the average result was reported in this study. A scanning electron microscope (SEM: JSM-6510, Jeol, Tokyo Japan) and energy dispersive spectroscopy (EDS: JED2300, Tokyo, Japan) were used to characterize the microstructure and wear tracks to shed some light on the wear mechanisms of the untreated and UNSM-treated vertical spindles that came into contact with two different V-belts.

All of the vertical spindles were washed before and after UNSM treatment using an ultrasonic bath (SD-80H, Mujigae, Seoul, Korea) in acetone (CH_3_)_2_CO for 10 min for deliverance of impurities. The surface roughness of the vertical spindle was measured using a two-dimensional surface profilometer (Mitutoyo: SJ-210, Kawasaki, Japan). The same profilometer was also used to obtain a surface profiling of the wear track formed on the surface of the V-belts. The surface hardness was measured using a portable hardness tester (GE Inspection Technologies: MIC 10, Lewistown, PA, USA). Both measurements were repeated at least 3 times in order to achieve a reliable result and the average result was reported in this study. A scanning electron microscope (SEM: JSM-6510, Jeol, Tokyo Japan) and energy dispersive spectroscopy (EDS: JED2300, Tokyo, Japan) were used to characterize the microstructure and wear tracks to shed some light on the wear mechanisms of the untreated and UNSM-treated vertical spindles that came into contact with two different V-belts.

## 3. Results and Discussion

### 3.1. Surface Roughness and Hardness

Surface roughness is an important surface property that affects the friction and wear behavior and performance of a component. Surface roughness is related to the number of asperities which affect the friction behavior by changing the true contact area of mating surfaces [19]. Figure 5a presents the comparison of arithmetical average surface roughness (*R_a_*) and mean values of ten peaks of surface roughness (*R_z_*) of the untreated and UNSM-treated vertical spindles. In this study, *R_a_* and *R_z_* parameters were selected to characterize the tribological behavior of vertical spindles because *R_a_* is the most widely known and it is the arithmetic mean deviation of the profile, but it does not describe contact surfaces entirely, since a completely different surface can have similar or even identical values of average surface roughness. Therefore, *R_z_* was provided to minimize the effect of unrepresentative asperities or valleys which occasionally occur and can give an erroneous value. The measurement results indicated that *R_a_* and *R_z_* were reduced after UNSM treatment from 1.723 to 0.998, corresponding to a 41.9% reduction, and from 10.773 to 5.733 µm, corresponding to a 46.6% reduction. The reduction in surface roughness after UNSM treatment may be attributed to the elimination of high peaks and valleys by introduced plastic deformation, resulting in reducing the number of asperities. In addition, a wave-like pattern can be formed on the surface of the UNSM-treated vertical spindle which is the result of forward and backward movement of a tip during UNSM treatment. It has been reported that, depending on the application purpose, the UNSM technique may increase or decrease the surface roughness of materials by optimizing the UNSM treatment parameters [10,13].

Besides surface roughness, the surface hardness also affects the friction and wear behavior of a component. Material with a harder surface is desirable to achieve a higher wear resistance. Figure 5b shows the comparison of surface hardness of the untreated and UNSM-treated vertical spindles. The UNSM-treated vertical spindle was found to have higher surface hardness compared to that of the untreated one, where the surface hardness was increased by 16.46%. The increase in surface hardness might be explained due to work hardening after SPD introduced by the application of UNSM treatment. In addition, it is also known that the UNSM treatment refines grain size, which leads to an increase in strength including hardness and also dislocation density [20,21]. In general, the relationship between hardness and grain size refinement can be explained by the Hall-Petch expression [22], where the smaller the grain size, the higher the hardness. Not only surface hardness, but also strength and ductility of ST3 may be enhanced by grain size refinement through the SPD method [23]. Practically, the effective depth of UNSM technique at RT for steels—in terms of hardened or grain size refined layer—may reach up to 200–300 µm depending on the UNSM treatment parameters, but that depth can be deepened even more by increasing the UNSM treatment temperature [13,24]. Hence, it can be assumed based on the percentage of hardness increase that the effective depth of the UNSM technique may reach up to over 100 µm.

### 3.2. Friction and Wear Behavior

In the belt transmission driving mechanism, the friction is one of the most important factors which determine the performance of the system [25]. Therefore, the tribological test is required to assess the influence of UNSM treatment to the vertical spindle by comparing it with the untreated one. Two different A and B belts were used as counterfaces to investigate the effect of different belts on the friction and wear behavior. Figure 6 shows the comparison of friction coefficient of the untreated and UNSM-treated vertical spindles that slid against A and B types of belts. The main difference between the A and B types of belts is the hardness and chemical composition, where the A type belt is harder/stiffer by about 18% compared to the B type belt due to the higher amount of C in the A type belt than that of the B type belt. The result showed that both the vertical spindles had similar running-in and steady-state friction behavior. At the beginning of the test, the friction coefficient of both the vertical spindles rapidly increased, where the friction coefficient of the untreated one reached 1.33, which is higher than the friction coefficient of the UNSM-treated one, which only reached 1.27. After reaching the peak, the friction coefficient of the untreated and UNSM-treated vertical spindles gradually reduced until reaching a steady-state friction behavior, where the friction coefficient of the UNSM-treated vertical spindles were found to be higher than that of the untreated ones. On the other hand, the friction coefficient of the untreated vertical spindles reduced with a steeper slope to a mean value of 0.95 and had a more stable steady-state friction, as shown in Figure 6a,b. In other words, the running-in and steady-state friction rate of the untreated vertical spindles was faster than that of the friction rate of the UNSM-treated ones. Figure 6a showed that the running-in friction coefficient of the untreated vertical spindle was higher than that of the UNSM-treated one, but the UNSM-treated one demonstrated a higher steady-state friction coefficient than that of the untreated one. However, as shown in Figure 6b, the running-in and steady-state friction coefficient of the untreated vertical spindle was lower than that of the UNSM-treated one. The friction coefficient of the untreated vertical spindle rapidly increased during running-in to a value of 2.75, and then gradually reduced and fluctuated to a value of about 2.20, while the friction coefficient of the UNSM-treated vertical spindle also rapidly increased during running-in to a value of about 3.13 and then gradually reduced and fluctuated to a value of about 2.68. Moreover, the friction coefficient of the UNSM-treated vertical spindle gradually reduced to a lower rate than that of the untreated one. The UNSM-treated vertical spindle had a higher friction coefficient and more stable friction behavior compared to the untreated one when rubbing against B belt. An increase in the friction coefficient of the UNSM-treated vertical spindle against both the belts in comparison with the untreated one is mainly attributed to the reduction in surface roughness and the presence of regular pattern formed on the surface of the UNSM-treated vertical spindles [16,26]. In addition, the distribution of the highest peaks over the contact area may also play an important role in controlling the running-in and steady-state friction behavior of the vertical spindles [27]. This is because the contact stress between the V-belt and sheave will result in elastic deformation of the V-belt into the asperities on the groove. The increased surface area interaction between two surfaces will result in an increase in the friction coefficient, while the wedging action is a result of the directional nature of the surface asperities in relation to the opposing V-belt surface [28]. The friction coefficient of a V-belt slip during transmission and slip determination was studied by Manin et. al. [29]. In the belt transmission system, the higher friction is more desirable because high friction means a better grip and seizure between belt and sheave, thereby resulting in a better efficiency and performance with no slip during transmission. The test results showed that the UNSM-treated vertical spindles demonstrated a higher friction coefficient in comparison with the untreated ones for both belts. Hence, it can be assumed that the UNSM-treated vertical spindle may deliver a better grip and seizure, and better efficiency with no slip in comparison with the untreated one. Generally, the application of the UNSM technique is required to minimize the friction coefficient of moving parts by modifying the surface integrity and mechanical properties [10,11,14], but in this study, the UNSM technique was able to increase the friction coefficient to avoid the absence of grip and seizure between belt and sheave to have better efficiency and performance with no slip during transmission, as the friction coefficient is one of the most important criteria when evaluating a cotton picker.

Friction between the mating surfaces of vertical spindle sheave and V-belt naturally will cause wear to emerge on both surfaces. V-belts as the counter surface were softer than the vertical spindle, hence the surface of the belt was more damaged and more visible compared to the surface of the vertical spindle. Therefore, the wear at surface of the V-belts was investigated to evaluate the effect of the UNSM-treated vertical spindle on the V-belt when compared to an untreated one. Figure 7a,b show the wear track formed on the surface of A belts after it was rubbed against both untreated and UNSM-treated vertical spindles. The surface profiles showed that the surface of V-belt that slid against UNSM-treated vertical spindle had a shallower wear than the V-belt that slid against the untreated one. The wear track that was formed on the surface of B belt showed different characteristics when compared to A ones. The wear track formed on the surface of B belts indicated that the wear had nearly similar width and depth for both counter surfaces of the untreated and UNSM-treated vertical spindles. To obtain a better understanding of the wear on both V-belts, the wear track was used to calculate the estimation of wear rate of both V-belts, as shown in Figure 8. Based on the calculation results, the wear rate of the A belt against untreated vertical spindle was 7.3 × 10^−4^ mm^3^/Nm and it decreased to 5.5 × 10^−4^ mm^3^/Nm when rubbed against UNSM-treated vertical spindle, corresponding to a 32.8% enhance in wear resistance. On the other hand, there was no significant change in the wear rate of the B belt against both untreated and UNSM-treated vertical spindles with a wear rate of 4.13 × 10^−4^ mm^3^/Nm and 4.05 × 10^−4^ mm^3^/Nm, respectively. The lower wear rate of the A belt against UNSM-treated vertical spindle may be attributed to the increase in surface hardness and also due to reduction in surface roughness which resulted in increasing the true contact area between two surfaces. The wear rate is proportional to number of contacting asperities which depend on the surface roughness [30]. In contrast with the B belt, the UNSM-treated vertical spindle indeed revealed higher friction compared to that of the A belt, but the wear rate was relatively same with the wear rate of B belt against untreated one. In general, the wear rate of the A belt against both untreated and UNSM-treated vertical spindles was higher than that of the B belt due to the different characteristics and mechanical properties of the V-belts [31].

### 3.3. Wear Mechanisms

The wear characteristics of V-belts as the counter surface of tribological tests were further investigated by analyzing SEM micrographs of the vertical spindles and V-belts to understand the wear mechanisms of both untreated and UNSM-treated vertical spindles against both A and B belts. Compared to the vertical spindles, the V-belts are relatively softer, therefore it is expected that more damage took place on the surface of the V-belt [32]. Figure 9 shows SEM micrographs of the wear tracks formed on the surface of A and B belts that slid against the untreated and UNSM-treated vertical spindles during the tribological test. The main difference between the A and B types of belts is the hardness and chemical composition, where the A type belt is harder/stiffer by about 18% compared to the B type belt due to the higher amount of C and ZnO in the A type belt than that of the B type one by about 28% and 40%, respectively. As for SEM micrographs of A belts, a similar damage can be seen at contact area of mating surface of A belt that slid against the untreated (Figure 9a,a1) and UNSM-treated (Figure 9b,b1) vertical spindles. The shear stress induced by friction force was sufficient to break the thread of the V-belt which is the cause of the main damage in this V-belt. In the case of UNSM-treated vertical spindle counter surface, a mild abrasive wear can be seen throughout the reciprocating sliding direction alongside the damaged V-belt thread. The mild abrasive wear can be found on the untreated vertical spindle counter surface as well, despite the fact that the wear was less visible, and the damaged wire also was more dominant on this surface. The higher friction produced by UNSM-treated vertical spindle may be the reason for more visible abrasive wear found on the UNSM-treated vertical spindles counter surface. Figure 9c,d show the SEM micrographs of the wear tracks formed on the surface of B belt that slid against the untreated and UNSM-treated vertical spindles. It can be seen for both cases; the abrasive wear can be easily observed alongside the reciprocating sliding direction due to the absence of wire in this V-belt. More scratches were formed on the surface of the untreated vertical spindle compared to that of the UNSM-treated one, despite having a slightly lower wear rate than that of the UNSM-treated one counter surface as shown in Figure 9. A similar abrasive wear also presented on both sides of the untreated vertical spindle counter surface, as shown in Figure 9c,c1. In the case of the UNSM-treated vertical spindle counter surface (see Figure 9d,d1), the surface looked less worn out than that of the untreated one counter surface, despite the fact that a deeper wear could also be found on some parts of the wear track. The smoother surface of the UNSM-treated vertical spindle may be the reason for less scratches and wear found on the UNSM-treated vertical spindle counter surface. SEM micrographs of the vertical spindles were also taken to investigate the effect of friction on both untreated and UNSM-treated vertical spindles. Figure 10a,c show the SEM micrographs of the vertical spindle against A and B belts, respectively. It can be seen that no wear could be found on the surface due to higher hardness of the vertical spindle compared to that of the V-belt. For the same reason, no wear could be observed on the surface of the UNSM-treated vertical spindle against A belt (Figure 10b). However, on the surface the UNSM-treated vertical spindle against B belt, some scratches could be observed as shown in Figure 10d. It appeared that the higher friction resulted from contact between the UNSM-treated vertical spindle and the B belt was the cause of the scratches.

EDS analysis was also performed on the V-belts after the tribological test in order to investigate the presence and level of oxidation layer on the wear tracks. Figure 11 shows ESD mapping and the oxide level (in wt.%) of A and B belts as counter surface of the untreated and UNSM-treated vertical spindles. It was found that the oxidation level in the A belt for the counter surface of the untreated vertical spindle was 29.97% and for the UNSM-treated one was 23.71%. Meanwhile, the oxidation level in B belt for counter surface of the untreated vertical spindle was 5.25% and for the UNSM-treated one was 5.96%. In case of the A belt, it appeared that the higher oxidation level on the counter surface of the untreated vertical spindle might show the presence of a thicker oxide layer compared to that of the UNSM-treated one counter surface, and results in a lower friction coefficient for the A belt against the untreated one [33]. Meanwhile, for the B belt, there was no significant change in oxidation level on both untreated and UNSM-treated vertical spindles’ counter surface. In general, a higher oxidation level was found in the A belt as counter surface of both vertical spindles might be the reason for lower friction compared to the B belt [34].

## 4. Conclusions

The UNSM technique was applied to the vertical spindle made of ST3 to improve the performance of the cotton picker by controlling the friction and wear behavior. It was assessed that the surface roughness of the UNSM-treated vertical spindle was reduced, while the surface hardness was increased in comparison with the untreated one. As for tribological performance, the reduced surface roughness of the UNSM-treated vertical spindle resulted in a higher friction coefficient compared to untreated one when tested against the V-belt as the counter surface. In case of wear resistance, unmeasurable wear occurred on the surface of the vertical spindle due to higher hardness of the vertical spindle compared to the V-belt. The V-belt as the counter surface of the vertical spindle wore out more severely than the vertical spindle. On the A belt, due to the higher surface hardness, the counter surface of the UNSM-treated vertical spindle had a lower wear rate than counter surface of the untreated one. However, on the B belt, despite the increase in surface hardness of the UNSM-treated vertical spindle, no significant difference in wear rate was found between counter surface of UNSM-treated and untreated vertical spindles. The higher fiction and wear resistance generated by the UNSM-treated vertical spindle are expected to improve the maintenance of the system in the application due to better grip and seizure between vertical spindle and the V-belt, resulting in better efficiency and performance with no slip during transmission.

## Figures and Tables

**Figure 1 materials-12-00773-f001:**
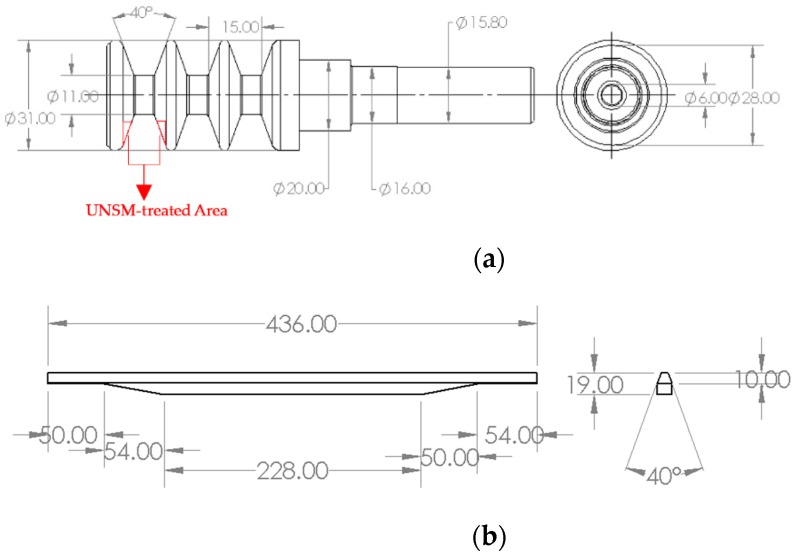
Nominal dimensions of vertical spindle (**a**) and V-belt (**b**) of a cotton picker. All dimensions in mm.

**Figure 2 materials-12-00773-f002:**
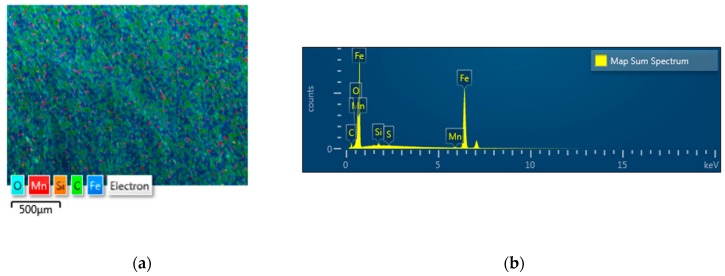
EDS mapping (**a**) along with distribution of chemical composition (**b**) of vertical spindle.

**Figure 3 materials-12-00773-f003:**
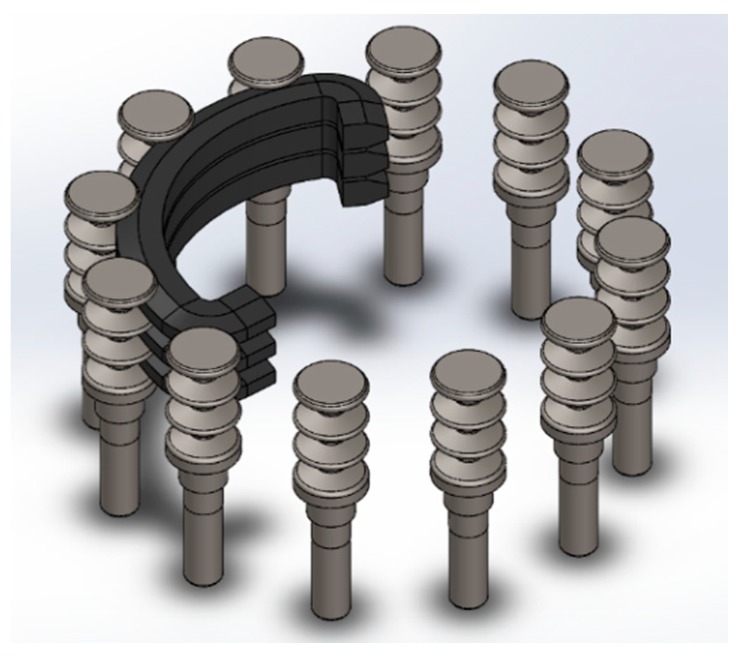
Working principle of a cotton picker.

**Figure 4 materials-12-00773-f004:**
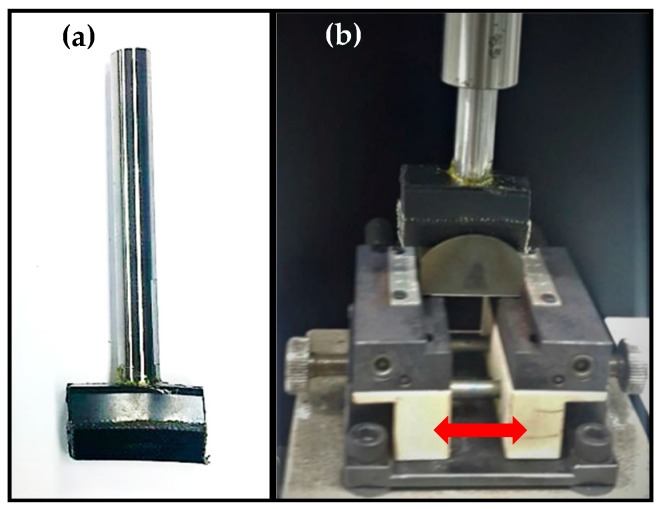
V-Belt fixation on the pin (**a**) and tribological test setting configuration (**b**).

**Figure 5 materials-12-00773-f005:**
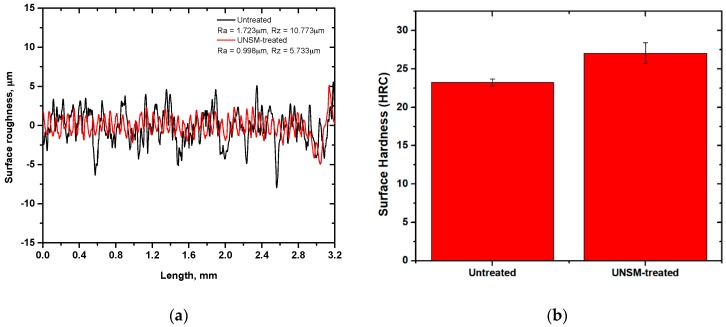
Comparison of surface roughness (**a**) and surface hardness (**b**) of the untreated and UNSM-treated vertical spindles.

**Figure 6 materials-12-00773-f006:**
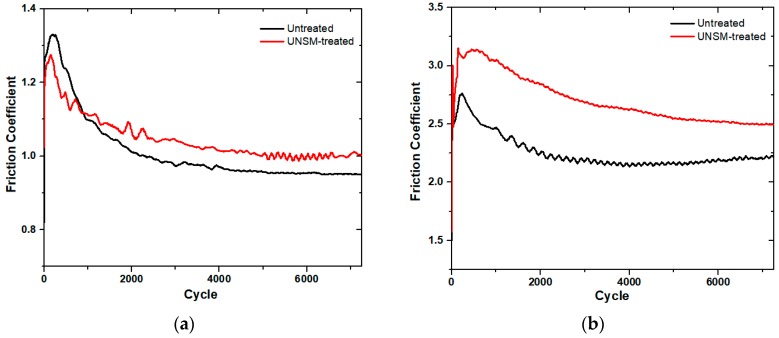
Comparison in the friction coefficient of the untreated and UNSM-treated vertical spindles slid against A (**a**) and B (**b**) belts.

**Figure 7 materials-12-00773-f007:**
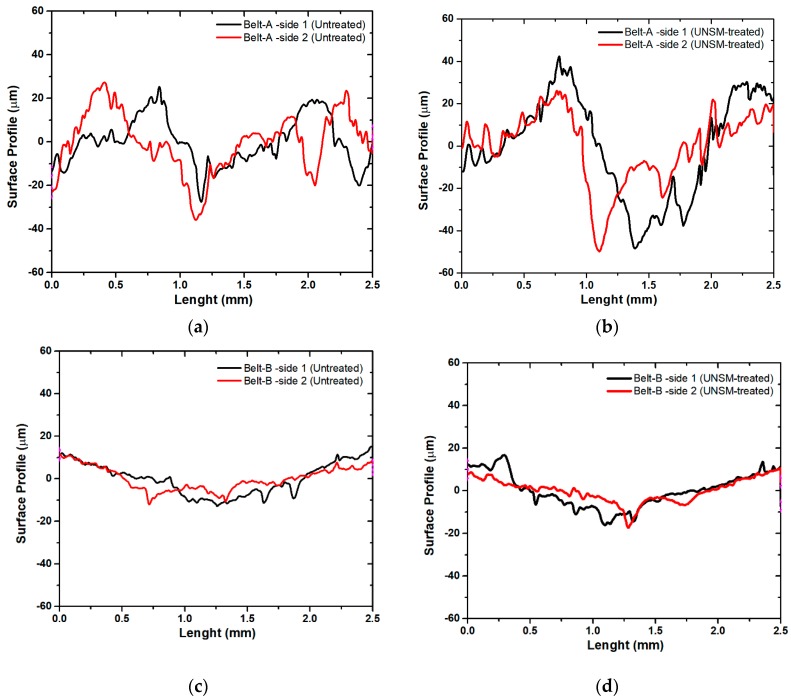
Comparison of wear track formed on the surface of A and B belts slid against untreated (**a**,**c**) and UNSM-treated (**b**,**d**) vertical spindles, respectively.

**Figure 8 materials-12-00773-f008:**
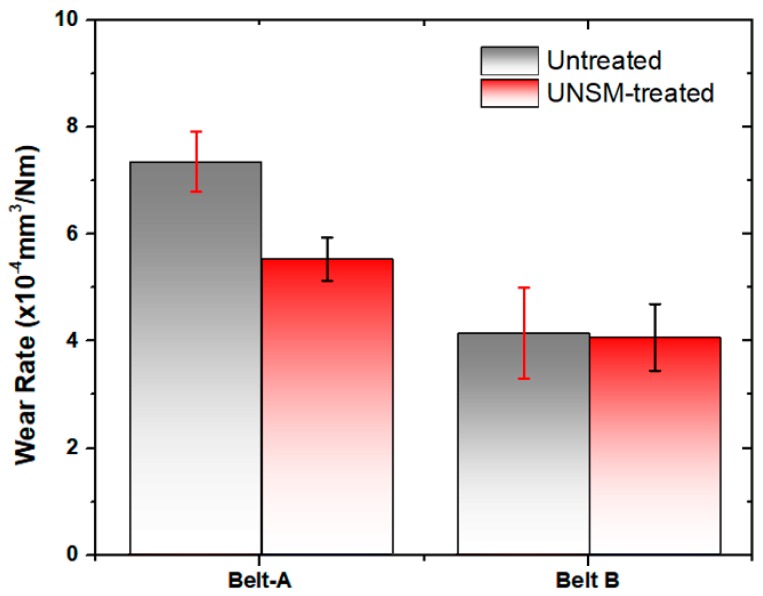
Comparison of wear rate of A and B belts slid against untreated and UNSM-treated vertical spindles.

**Figure 9 materials-12-00773-f009:**
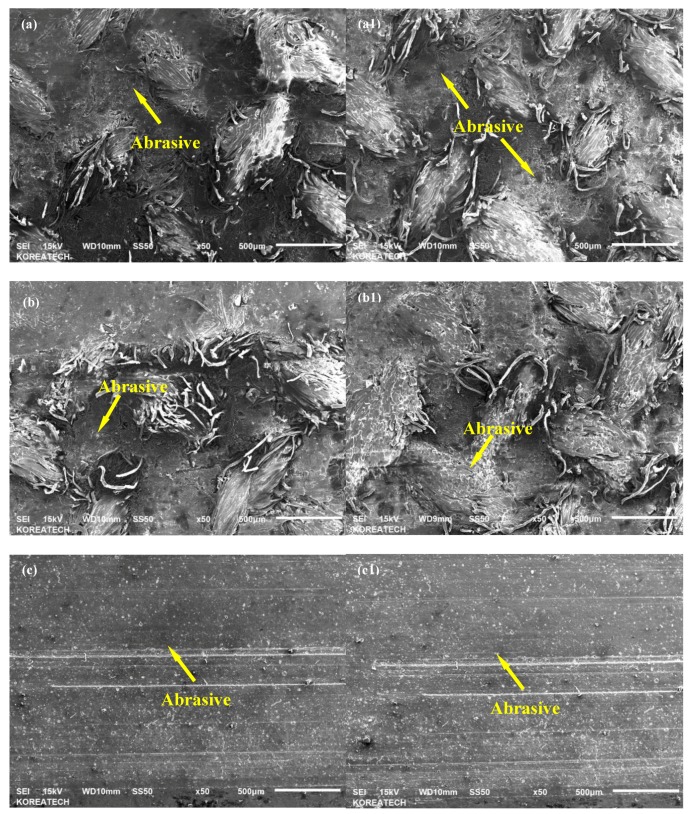
SEM micrographs of wear track of A belt as counter surface of untreated (**a**,**a1**) and UNSM-treated (**b**,**b1**) vertical spindles, wear track of B belt as counter surface of untreated (**c**,**c1**) and UNSM-treated (**d**,**d1**) vertical spindles.

**Figure 10 materials-12-00773-f010:**
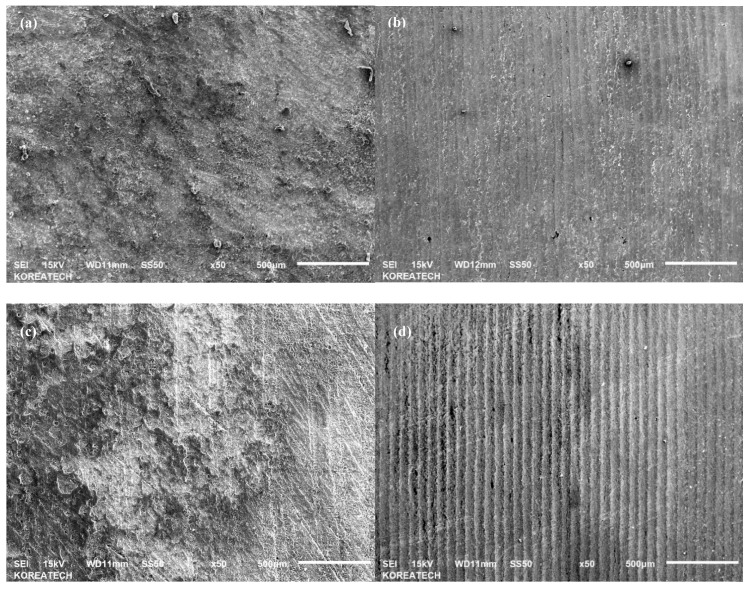
SEM micrographs of untreated vertical spindles against A belt (**a**) and B belt (**c**), and UNSM-treated vertical spindles against A belt (**b**) and B belt (**d**).

**Figure 11 materials-12-00773-f011:**
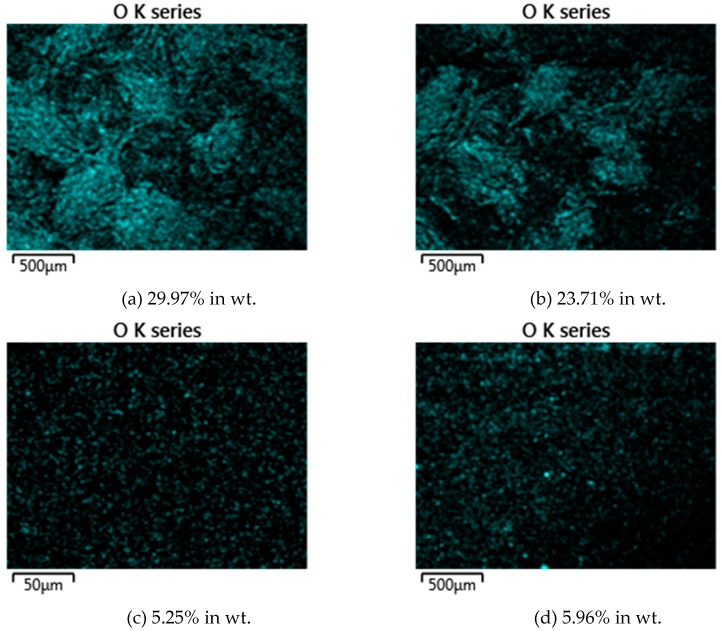
EDS mappings of the A belt against untreated (**a**) and UNSM-treated (**b**), and B belt against untreated (**c**) and UNSM-treated (**d**) vertical spindles showing the oxidation level generated during reciprocating sliding.

**Table 1 materials-12-00773-t001:** Shore hardness, chemical composition (in wt.%) and types of two different belts.

Types	Shore Hardness, A	Natural Rubber (C_5_H_8_)	C	ZnO	S
Rubber A	55	69	25	5	1
Rubber B	45	78	18	3	1

**Table 2 materials-12-00773-t002:** Mechanical properties of low-carbon steel (ST3).

**Low-Carbon Steel (ST3)**	**Tensile Strengtd, MPa**	**Yield Strengtd, MPa**	**Elastic Modulus, GPa**	**Shear Modulus, GPa**	**Poisson’s Ratio**	**Elongation, %**
	>490	>290	190	73	0.29	>21%

**Table 3 materials-12-00773-t003:** Chemical composition of low-carbon steel (ST3) in wt.%.

**Low-Carbon Steel (ST3)**	**C**	**Si**	**Mn**	**Ni**	**S**	**P**	**Cr**	**Cu**	**As**	**Fe**
	0.32	0.24	0.60	0.25	0.04	0.035	0.25	0.25	0.08	97.0

**Table 4 materials-12-00773-t004:** Optimized treatment parameters of the UNSM technique.

Frequency, kHz	Amplitude, µm	Speed, mm/min	Static Load, N	Feed-Rate, µm	Ball Diameter, mm
20	20	2000	20	30	2.38

**Table 5 materials-12-00773-t005:** Tribological test conditions.

Normal Load, N	Reciprocating Cycles	Linear Speed, cm/s	Stroke, mm	Temperature, °C	Contact Stress, Pa
5.0	7250	2.51	4.0	>25.0	245.8

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
