# Peer review of "Experimental Investigation on Friction and Wear Behavior of the Vertical Spindle and V-belt of a Cotton Picker"

_materials, 2019, doi:10.3390/ma12050773_

Reviewer 1 Report

line 18, 35: 'the vertical spindle'

line 38: 'the main shaft'

line 62: 'was treated by'

line 64: 'the vertical spindle'; 'that comes'

line 78: 'careless sample storage'

The composition and material of the V-belts must be described in more detail (type of rubber, use of fabrics or not?, ...), not only the shore hardness.

line 99: 'utilize ultrasonic'

line 102: 'balls are used' - or was there only one?

line 103: you mention optimized treatment parameters. To which ctriteria optimized?

chapter 2.3.: you should provide at least one drawing/sketch/picture to describe the working principle of the cotton picker. With only a verbal description someone not familiar with this device is lost.

line 197-200: you mention the effective depth of USNM. How deep reached the USNM treatment in your case - the spindles. Please provide some measurements/micrograph/... to answer this question.

line 212: repeat the main difference(s) between the A and the B belt here. Thus it is easier for the reader to follow the discussion; also in the caption of Fig. 6.

line 295 ff: for the discussion of Fig. 9 more details on the material and composition of the V-belts are necessary and should be included here.

line 319: 'the vertical spindle'

line 336: 'EDS'

line 359: 'The V-belt as the'

line 362: what do you mean with 'despite the increase in surface hardness'? Of the belt? ...

Author Response

Ref.: Materials-453400R1

Title: Experimental Investigation on Friction and Wear Behavior of the Vertical Spindle and V-belt of Cotton Picker 

Authors: A. Amanov J.P.B.A. Sembiring and T. Amanov 

We would like to thank the anonymous reviewers for the time and effort to review the manuscript and provide us with valuable comments and suggestions, which helped us to revise the manuscript. Most comments seemed to be fair and reasonable, so we paid heed to most of their advice and suggestions, and the manuscript has been revised according to all the comments to the best of our ability. We believe that the contents and the clarity of our manuscript are much improved in the revised version. Below are point-by-point responses for the two anonymous reviewers` comments: 

Responses to comments

Reviewer #1:

line 18, 35: 'the vertical spindle'

RESPONSE: The corrections have been made in the revised manuscript.

line 38: 'the main shaft'

RESPONSE: The correction has been made in the revised manuscript.

line 62: 'was treated by'

RESPONSE: The correction has been made in the revised manuscript.

line 64: 'the vertical spindle'; 'that comes'

RESPONSE: The corrections have been made in the revised manuscript.

line 78: 'careless sample storage'

RESPONSE: The correction has been made in the revised manuscript.

The composition and material of the V-belts must be described in more detail (type of rubber, use of fabrics or not?, ...), not only the shore hardness. 

RESPONSE: The shore hardness, chemical composition and types of two different belts are listed in Table 1 in the revised manuscript. Construction of both belt materials is rubber and both are not reinforced.

line 99: 'utilize ultrasonic'

RESPONSE: The correction has been made in the revised manuscript.

line 102: 'balls are used' - or was there only one?

RESPONSE: The correction has been made in the revised manuscript. A new ball was used for each test.

line 103: you mention optimized treatment parameters. To which ctriteria optimized?

RESPONSE: The optimization of UNSM technique parameters was made based on the reduction in surface roughness and increase surface hardness.

chapter 2.3.: you should provide at least one drawing/sketch/picture to describe the working principle of the cotton picker. With only a verbal description someone not familiar with this device is lost.

RESPONSE: Actually, Fig. 3 describes the working principle of the combination of belt and spindle of the cotton picker.

line 197-200: you mention the effective depth of USNM. How deep reached the USNM treatment in your case - the spindles. Please provide some measurements/micrograph/... to answer this question.

RESPONSE: As it mentioned in the article, the effective depth of UNSM technique at RT for steels may reach up to 200-300 µm depending on the UNSM treatment parameters. Hence, it can be assumed based on the percentage of hardness increase by about 17% that the effective depth of UNSM technique may reach up to over 100 µm.

line 212: repeat the main difference(s) between the A and the B belt here. Thus it is easier for the reader to follow the discussion; also in the caption of Fig. 6.

RESPONSE: The main difference between the A and B types of belts is the hardness and chemical composition, where A type belt is harder/stiffer by about 18% compared to the B type belt due to the higher amount of C in A type belt than that of the B type belt.

line 295 ff: for the discussion of Fig. 9 more details on the material and composition of the V-belts are necessary and should be included here.

RESPONSE: The main difference between the A and B types of belts is the hardness and chemical composition, where A type belt is harder/stiffer by about 18% compared to the B type belt due to the higher amount of C and ZnO in A type belt than that of the B type one by about 28 and 40%, respectively.

line 319: 'the vertical spindle'

RESPONSE: The correction has been made in the revised manuscript.

line 336: 'EDS'

RESPONSE: The correction has been made in the revised manuscript.

line 359: 'The V-belt as the'

RESPONSE: The correction has been made in the revised manuscript.

line 362: what do you mean with 'despite the increase in surface hardness'? Of the belt? ...

RESPONSE: 'despite the increase in surface hardness of the UNSM-treated vertical spindle' has been corrected. The surface hardness of the belt has not been measured in this study.

Reviewer 2 Report

This article focuses on experimental investigation of friction and wear behaviour of spindle and V-belt. The topic of this manuscript falls within the scope of Materials journal.

The literature review included in Introduction section gives a detailed and properly structured description of the problem, with mentioning of main methods to improve the performance of cotton pickers.

The importance and the number of references cited are satisfactory. The most of references cited were published in 21st-century.

In my opinion the experimental methods are suitable and envious. The investigations have been well designed. The presentation of the results is quite clear. The quality of communication is acceptable. 

I feel that due to the specific topic of this article the scientific soundness of this manuscript is limited to the small group of researches and technicians in textile industry. The structure of this paper and the quality of figures are very good.

I would like to recommend this paper for publication in the “Materials” journal after solving some minor  comments:

-  Why only Ra and Rz surface roughness parameters were selected to characterize the tribological behaviour of vertical spindles?

-   The method of determination the friction coefficient value should be described in detail. Was this the direct or indirect method?

Author Response

MATERIALS

Ref.: Materials-453400R1

Title: Experimental Investigation on Friction and Wear Behavior of the Vertical Spindle and V-belt of Cotton Picker

Authors: A. Amanov J.P.B.A. Sembiring and T. Amanov 

We would like to thank the anonymous reviewers for the time and effort to review the manuscript and provide us with valuable comments and suggestions, which helped us to revise the manuscript. Most comments seemed to be fair and reasonable, so we paid heed to most of their advice and suggestions, and the manuscript has been revised according to all the comments to the best of our ability. We believe that the contents and the clarity of our manuscript are much improved in the revised version. Below are point-by-point responses for the two anonymous reviewers` comments: 

Responses to comments

Reviewer #3:

This article focuses on experimental investigation of friction and wear behaviour of spindle and V-belt. The topic of this manuscript falls within the scope of Materials journal.

The literature review included in Introduction section gives a detailed and properly structured description of the problem, with mentioning of main methods to improve the performance of cotton pickers.

The importance and the number of references cited are satisfactory. The most of references cited were published in 21st-century.

In my opinion the experimental methods are suitable and envious. The investigations have been well designed. The presentation of the results is quite clear. The quality of communication is acceptable. 

I feel that due to the specific topic of this article the scientific soundness of this manuscript is limited to the small group of researches and technicians in textile industry. The structure of this paper and the quality of figures are very good.

I would like to recommend this paper for publication in the “Materials” journal after solving some minor comments:

- Why only Ra and Rz surface roughness parameters were selected to characterize the tribological behaviour of vertical spindles?

RESPONSE: The authors selected only Ra and Rz surface roughness parameters to characterize the tribological behaviour of vertical spindles because the most widely known and used roughness parameter is the arithmetic mean deviation of the profile Ra; however it does not describe contact surfaces entirely, since a completely different surface can have similar or even identical value of average surface roughness. Also, Ra provides a very good overall description of height variations, but it does not enough information on wavelengths and it is not sensitive to small changes in profile after UNSM. Rz is defined as the distance between the averages of five highest asperities and the five lowest valleys. The reason for taking and average value of asperities and valleys is to minimize the effect of unrepresentative asperities or valleys which occasionally occur and can give an erroneous value if taken singly. Rz is more reproducible and advocated by ISO standard. In many tribological applications, height of the highest asperities above the mean line is an important parameter because damage may be done to the interface by the few high asperities present on one of two surfaces.

-  The method of determination the friction coefficient value should be described in detail. Was this the direct or indirect method?

RESPONSE: The friction coefficient was obtained as the ratio of tangential force (F) to the normal load (N), where both were measured using a sensor. This is the direct method.